

# Using sentinel nodes to evaluate changing connectivity in a protected area network

Paul O'Brien[1], Natasha Carr[2] and Jeff Bowman[1]

[1] Ontario Ministry of Natural Resources and Forestry (MNRF), Peterborough, Ontario, Canada
[2] Ontario Ministry of the Environment, Conservation and Parks (MECP), Peterborough, Ontario, Canada

## ABSTRACT

It has been recognized that well-connected networks of protected areas are needed to halt the continued loss of global biodiversity. The recently signed Kunming-Montreal biodiversity agreement commits countries to protecting 30% of terrestrial lands in well-connected networks of protected areas by 2030. To meet these ambitious targets, land-use planners and conservation practitioners will require tools to identify areas important for connectivity and track future changes. In this study we present methods using circuit theoretic models with a subset of sentinel park nodes to evaluate connectivity for a protected areas network. We assigned a lower cost to natural areas within protected areas, under the assumption that animal movement within parks should be less costly given the regulation of activities. We found that by using mean pairwise effective resistance (MPER) as an indicator of overall network connectivity, we were able to detect changes in a parks network in response to simulated land-use changes. As expected, MPER increased with the addition of high-cost developments and decreased with the addition of new, low-cost protected areas. We tested our sentinel node method by evaluating connectivity for the protected area network in the province of Ontario, Canada. We also calculated a node isolation index, which highlighted differences in protected area connectivity between the north and the south of the province. Our method can help provide protected areas ecologists and planners with baseline estimates of connectivity for a given protected area network and an indicator that can be used to track changes in connectivity in the future.

## INTRODUCTION

Protected and conserved areas are considered a fundamental conservation strategy for protecting biodiversity (*McNeely, 1994*; *Naughton-Treves, Holland & Brandon, 2005*; *Watson et al., 2014*). With proper placement and management, protected areas can reduce species extinction rates, habitat destruction, and hunting mortality (*Andam et al., 2008*; *Butchart et al., 2012*; *Hilborn et al., 2006*), while also having the potential to secure valuable ecosystem services (*Mitchell et al., 2021*; *Naidoo et al., 2008*; *O'Brien et al., 2023*) and benefit human wellbeing (*Naidoo et al., 2019*). Nevertheless, recent reviews have called to question the effectiveness of current protected areas at reducing human pressure and

Corresponding author
Paul O'Brien,
obrienp1@myumanitoba.ca

biodiversity loss (*Geldmann et al., 2019*; *Maxwell et al., 2020*). Indeed, biodiversity inside and outside of park boundaries is increasingly threatened by accelerated human impacts including development, hunting pressure, and recreation (*Barrueto et al., 2022*; *Craigie et al., 2010*; *Laurance et al., 2012*; *Tucker et al., 2018*). Further, individual protected areas are often too small and isolated to support populations of large or vagile animals (*Williams, Rondinini & Tilman, 2022*). Habitat loss, fragmentation, and degradation continue to be major threats to biodiversity loss through reductions in functional connectivity (*Goodwin & Fahrig, 2002*; *Haddad et al., 2015*) and can lead to isolation of populations within protected areas (*Sawaya, Clevenger & Schwartz, 2019*). While protected areas can help to safeguard biodiversity (*Gray et al., 2016*), human impact outside of parks can inhibit species ability to move (*Tucker et al., 2018*) and many species are expected to experience shifts in the location of suitable habitat with climate change, necessitating the need to move (*Bellard et al., 2012*; *Chen et al., 2011*; *Parmesan, 2006*). Therefore, maintaining and restoring connectivity among protected areas is a high priority for the protection of biodiversity within protected areas (*Newmark et al., 2023*).

Landscape connectivity is the degree to which landscapes facilitate or impede movement of organisms among resource patches (*Taylor et al., 1993*; *Tischendorf & Fahrig, 2000*). Connectivity is critical for species movement and gene flow through dispersal and migratory movements (*Noss et al., 2012*). Consequently, loss of connectivity can be a driver of species extinctions (*Hooftman, Edwards & Bullock, 2016*; *Thompson, Rayfield & Gonzalez, 2017*) as isolated populations face increased risk of extinction due to inbreeding depression, stochastic events, and reduced opportunity for genetic rescue (*Hoffmann, Miller & Weeks, 2021*; *Pimm, Dollar & Bass, 2006*). Compared to isolated core areas, ecological networks, that is, networks of core areas (*e.g.*, protected areas, OECMs, and unprotected Key Biodiversity Areas) and corridors are better able to connect populations, maintain ecosystem function, and support species persistence in the face of climate change and landscape modification (*Heller & Zavaleta, 2009*; *Hilty et al., 2020*; *Schloss et al., 2022*). Connectivity is therefore thought to be essential for the long-term persistence of biodiversity (*Ward et al., 2020*). While the need to build well-connected systems of protected areas has been identified as important by previous international biodiversity agreements (Aichi Targets) and more recently reinforced in the COP15 biodiversity agreement (Goal A and Target 3 of Kunming-Montreal Global Biodiversity Agreement; *Convention on Biological Diversity, 2022*), connectivity is often not incorporated into conservation planning (*Carroll & Ray, 2021*; *Maxwell et al., 2020*). To better address connectivity in conservation strategies, and more specifically into parks development and planning, we suggest that land-use planners and decision-makers across various jurisdictions (local to national) require tools to (1) map contemporary connectivity among protected areas to guide identification of new areas for protection or restoration; and (2) evaluate the effects of future land-use changes or conservation interventions on connectivity of the protected area network.

Despite the rarity of connectivity being integrated into protected areas planning, a wide variety of techniques exist for assessing protected area connectivity. Circuit-theoretic

models of connectivity are one such method that has gained increasing popularity. These models, which draw on the analogy between animals moving through a landscape and electrical current moving through a circuit, allow for the identification of multiple movement pathways between defined source and destination nodes (*McRae et al., 2008*). Typically, these nodes have represented core habitat patches that animals are likely to move among, often including protected areas. Many studies have used a circuit-theoretic approach with Circuitscape software to model connectivity among protected areas at various scales, including regionally (*Belote et al., 2016*; *Dickson et al., 2017*), on a continental scale (*Barnett & Belote, 2021*), and globally (*Brennan et al., 2022*). Using just two inputs, a cost-to-movement surface and node locations, one output of Circuitscape is a cumulative current density map where current density, measured in amperes, is proportional to the probability of an animal using any given pixel during a random walk through the landscape. The resulting current density map can be used to identify areas that contribute importantly to connectivity between protected areas or other habitat patches or likewise, areas where connectivity is poor and could be restored. Circuit theoretic models provide a suitable framework for examining contemporary protected area connectivity, which satisfies our first suggested requirement of tools for incorporating connectivity into conservation planning. The second requirement is being able to track future changes in connectivity through time, which may be achieved with use of a connectivity indicator.

Indicators are goal-specific metrics that can be measured to determine whether objectives are being achieved. In the context of connectivity, many different metrics have been developed (see *Keeley, Beier & Jenness, 2021* for a detailed review), however minimal progress has been made towards integrating these metrics into on-the-ground action (*Theobald et al., 2022*). Building on work by *Jaeger (2000)*, *Theobald et al. (2022)* describe the desirable properties of a connectivity indicator including that it should reflect within- and between-patch connectivity, be computationally efficient, and be simple to measure and communicate. To this list, we add that a connectivity indicator should be repeatable to track progress in connectivity goals through time. *Jaeger (2000)* introduced a quantitative measure of landscape fragmentation, the effective mesh size ($M_{eff}$), which can be interpreted as the probability of two animals, placed in different locations in a study area, encountering each other in the same patch. The effective mesh size satisfies many of the desirable properties of a connectivity metric and can be modified to incorporate within-patch connectivity (*Deslauriers et al., 2018*; *Spanowicz & Jaeger, 2019*); however, it may not be suitable for an analysis specific to protected areas and their surroundings, since $M_{eff}$ would exclude unprotected natural areas within the matrix from measurements of connectivity. A more appropriate indicator of protected area connectivity may be provided directly from Circuitscape.

Pairwise effective resistance is a second output of circuit theory models, in addition to current density, and is a measure of the cost of moving between two nodes or habitat patches, calculated for all pairs of nodes. The more potential pathways there are between two nodes, the lower the effective resistance will be. *Brennan et al. (2022)* used a version of

effective resistance, which they called the 'Protected Area Isolation' (PAI) index, to measure connectivity for a global protected area network. The PAI index calculates the cost of movement between a focal protected area and all other protected areas in the network, and so reflects the degree of isolation of the focal park. This metric adds to existing global connectivity metrics by incorporating functional connectivity and can be implemented efficiently at a global scale. PAI may be less suitable when the objective is to track network connectivity through time however, given that calculation of the PAI index is dependent on the membership of the contemporaneous protected area network, which may change over time. Further, because the number of pixels substantially increases at finer resolutions (<1 km), which may be more useful to regional planners, calculation of the PAI index may not be as computationally efficient owing to the need to run Circuitscape twice—once to produce the current density map and again to calculate the PAI index using a different mode (*Brennan et al., 2022*). We suggest an alternative would be to use the mean pairwise effective resistance (MPER) as an indicator of connectivity. This is the mean of all pairwise effective resistances and reflects a measure of overall network connectivity. Pairwise effective resistance can also be used to calculate isolation values of individual network nodes, similar to *Brennan et al.*'s *(2022)* PAI index. Further, both pairwise effective resistance and a cumulative current density map are produced within a single run of pairwise mode Circuitscape, making this a more efficient method.

One consideration with our proposed MPER indicator, and one that arises with other connectivity indicators, is that they are not necessarily repeatable across time if the set of network nodes is dynamic. That is, as the network changes through addition or loss of protected areas, the set of nodes changes and so recalculations of the indicator across time are no longer comparable (*i.e.*, there is a conflation between space and time). To deal with this problem, we propose a modified version of park-to-park connectivity methods whereby we model connectivity using a set of sentinel nodes. We define sentinel nodes as a subset of protected areas used to assess connectivity of the full protected area network over time. This fixed set of nodes provides a repeatable framework for modelling connectivity and tracking changes in connectivity of a network through time. To develop the idea of a sentinel node indicator, we first carried out sensitivity analyses on smaller, but real-world landscapes. Next, we tested the ability of our method at a larger scale by modelling connectivity of the protected areas network in the province of Ontario, Canada.

The current protected and conserved area network in Ontario consists of >1,400 protected areas covering 10.9% of Ontario (*Environment and Climate Change Canada, 2023*) and accounting for 1.17% of the national coverage (*Environment and Climate Change Canada, 2023*). Canada has committed to protecting 30% of terrestrial lands in a well-connected network of protected areas by 2030. The current state of connectivity of the Ontario protected area network is unknown. Our assessment will help to provide a baseline understanding of park-to-park connectivity for the province's protected areas, and an indicator of parks connectivity for future use. Our analysis will also support decisions required to achieve ambitious national targets. We note that portions of this text were previously published as part of a preprint (https://www.biorxiv.org/content/10.1101/2023.04.25.538164v1.full.pdf).
**Table 1  List of cost values used to classify the cost surface and types of landscape features assigned each cost rank.**

| Cost value | Landscape features |
|---|---|
| 0.1 | Natural areas* within protected areas boundaries |
| 1 | Natural areas* outside of protected areas boundaries |
| 10 | Minor roads, pasturelands, forestry (cuts < 35 yrs old) |
| 100 | Croplands, two-lane highways |
| 1,000 | Cities, railroads, multi-lane highways, lakes ($\geqq$ 10 ha), rivers (flow > 28 m$^3$/s), mines, dams, nighttime lights |

**Notes:**
*Natural areas refers to all terrestrial, non-anthropogenic landscape features including forests, wetlands, grasslands, *etc.*
See *Pither et al. (2023)* for a detailed description of landscape feature classifications and data layers used.

## METHODS

### Cost surface and nodes

We modelled landscape connectivity using a circuit theoretic approach, which takes advantage of the analogy between electricity moving through a circuit and animals moving across a landscape, allowing for the identification of multiple movement pathways. Electricity travels through a circuit according to a random walk, and animals often move using a random walk, so circuits can be used to model these movements (*Doyle & Snell, 1984*). Modelling connectivity using circuit theory requires two inputs: a cost surface and a file with the location of source and destination nodes. A movement cost surface represents features in a landscape (*e.g.*, roads, rivers, forests, *etc.*) by the degree to which they facilitate or impede movement (*McRae et al., 2008*). For the following analyses, we used the 300-m resolution cost surface of Canada produced by *Pither et al. (2023)*, which classifies the landscape according to a cost scheme using four values including 1 (low cost), 10, 100, and 1,000 (high cost) (Table 1). *Bowman et al. (2020)* showed that current density maps are not sensitive to the absolute costs of a cost surface provided that costs are in the correct rank order. *Pither et al. (2023)* modelled landscape connectivity for terrestrial non-volant fauna, such that natural features like forests, wetlands, and grasslands were assigned a low cost, while anthropogenic features like roads and cities, as well as large water bodies and mountains were represented with a high cost to movement. This approach, which has been used in many other studies, models landscape connectivity based on degree of naturalness or human modification and makes the assumption that more natural landscapes are less costly for many animals to move through and better facilitate ecological processes (*Krosby et al., 2015*; *Spencer et al., 2010*; *Theobald et al., 2012*). While these upstream, naturalness models do not measure true functional connectivity, they have been found to predict areas important for functional connectivity of multiple species (*Koen et al., 2014*; *Pither et al., 2023*; *Wood et al., 2022*).

*Pither et al. (2023)* built their cost surface using the most up-to-date spatial data layers including the Canadian Human Footprint (*Hirsh-Pearson et al., 2022*) and an updated national road layer (*Poley et al., 2022*). We modified the 300-m cost surface of *Pither et al. (2023)* by adding a fifth lowest cost, which we assigned to natural areas within protected area boundaries under the assumption that these areas are less costly to move through than natural areas outside park boundaries (*Spencer et al., 2010*; Table 1). While we do not

propose that the natural landscapes themselves would differ within *vs* outside park boundaries, we reasoned that regulation of various human activities, such as resource extraction, new hydro development, hunting, and recreation, within protected areas as well as traditional land use practices and stewardship within Indigenous Protected and Conserved Areas (IPCAs) can provide benefits through decreased disturbance and mortality, allowing wildlife to move more easily through protected and conserved areas (*Fryxell et al., 2020*; *Hebblewhite & Whittington, 2020*; *Indigenous Circle of Experts, 2018*; *Obbard et al., 2017*; *Schuster et al., 2019*). We also note that a similar approach was taken by *Spencer et al. (2010)* who weighted pixels in protected areas according to their degree of regulation. All higher cost (10, 100, and 1,000) land cover features within park boundaries, such as roads, developments, and water retained their original costs given they still impede movement within parks. With protected areas having the lowest cost on the landscape, adding a new park in the future, even if it is not a network node, will reduce the overall mean pairwise effective resistance relative to the network prior to addition of the park. We considered this to be an important property of an indicator for a protected area network.

The second input needed is a node file identifying the source and destination locations between which to evaluate connectivity. Often, source and destination nodes have been defined using *a priori* knowledge about core habitat patches that animals are most likely to move between and in the case of a park-to-park analysis, the parks serve as the nodes. Using all protected areas within a study area can be computationally demanding, especially for large landscapes at a high resolution. To help deal with this, many studies set criteria for parks to include, often limiting to parks above a certain size (*Barnett & Belote, 2021*; *Brennan et al., 2022*; *Dickson et al., 2017*). This can greatly reduce the number of park nodes included, thus reducing computation time, however it can result in potentially biased current density estimates based on the spatial distribution of nodes (*Koen et al., 2014*). More recently, omnidirectional methods have been developed which shift the nodes outside of the study area and are useful in cases where source and destination locations are unknown or for modelling landscape connectivity without reference to specific node locations (*Koen et al., 2014*; *Phillips et al., 2021*; *Pither et al., 2023*). Omnidirectional connectivity models also facilitate experimentation to determine the sufficient number and placement of nodes for accurate estimates (*Koen et al., 2014*) and allow reduced bias due to node placement by removing nodes from inside the study area. Placing nodes within park boundaries is the most appropriate method for evaluating park-to-park connectivity; however, omnidirectional methods allow nodes to be independent of the parks network and therefore estimates of connectivity are repeatable over time, even as changes are made to the underlying distribution and abundance of protected areas in the network.

To evaluate connectivity of a protected area network over time, we draw on properties of both park-to-park and omnidirectional methods to achieve our objective. We developed the sentinel node method, which involves randomly selecting a subset of parks to represent the full parks network. We propose that by using a sufficient subset of nodes, we should be able to accurately evaluate connectivity for a full protected area network. We use a single pixel placed within the park boundaries to represent a sentinel node. This allows an

estimation of both internal park connectivity and connectivity between parks. Internal park connectivity should be important for more fine-scale, daily movements of animals and species that have home ranges contained fully within park boundaries, whereas interpark connectivity would be most important for dispersal or migratory movements, species with large home ranges (*i.e.*, larger than park boundaries), and for potential range expansions and contractions. Further, the consistent use of a randomly selected, fixed subset of park nodes allows changes in connectivity to be compared over time.

The following analyses were all performed using the Julia implementation of Circuitscape in pairwise mode (Julia version 1.7.3; *Hall et al., 2021*). Circuitscape provides two outputs which can be used to evaluate connectivity of a protected areas network. The first, a current density map, represents the probability of movement across the landscape and can be used to identify critical connectivity areas within or between protected areas. The second, pairwise effective resistance, is a measure of the cost of travelling between two pairs of park nodes for all pairs of nodes. The more potential pathways there are between nodes, the lower the effective resistance will be. We suggest the mean of the pairwise effective resistance (MPER) can serve as an indicator of overall network connectivity that can be tracked over time.

## Representation of a full network and sensitivity to changes in the network

To determine how well the sentinel node method captures connectivity of the full park network, we used a small study area to model connectivity for a series of random park subsets and compared these to estimates of the full 'true' network from the same small area. We tested these methods in a ~57,000 km$^2$ area in eastern Ontario that represents 1 of 5 zones used to manage parks in the province. Using parks from the Canadian Protected and Conserved Areas Database (*Environment and Climate Change Canada, 2023*), which included provincial, federal, and private protected areas, we generated three random subsets of 20 parks in this small study area to represent three possible sentinel node scenarios. We then combined these three subsets, removing duplicates, to represent the full 'true' network, resulting in 52 parks. Park centroids were used as the location for nodes as opposed to the full park polygons in order to capture variability in connectivity within park boundaries (*Brennan et al., 2022*). For this analysis, all raster cells classified as natural cover (cost = 1) within park boundaries (both node parks and non-node parks) were re-assigned the lower cost of 0.1, representing the lowest cost on the landscape. All other cost values within park boundaries remained the same as the costs used by *Pither et al. (2023)* (*i.e.*, 10, 100, or 1,000). Current density estimates of the three subsets and full network were compared using Spearman rank correlations of 1,000 random pixel values.

We tested how well the sentinel node method could detect future changes in the network using six simulated scenarios in the small study area—three development scenarios that would reduce connectivity and three park-addition scenarios that would enhance connectivity. For the development scenarios, we first added a 157-km$^2$ high-cost development (cost = 1,000) in an area of relatively low cost with a high-cost road feature connecting it to an existing highway. In the second scenario, we added a 199-km$^2$

development, in addition to the previous development and in the third scenario we added a 676-km$^2$ development to the previous two. For the park-addition scenarios we (1) added a new 33-km$^2$ protected area (cost = 0.1 for natural areas) to a heavily developed region on the landscape, (2) added a second 336-km$^2$ protected area in addition to the first (cost = 0.1 for natural areas), and (3) added a third 150-km$^2$ protected area. We ran each of the six described scenarios for the three random sentinel node networks and the full 'true' network. To measure potential changes in connectivity with each of the new development scenarios, we calculated mean pairwise effective resistance (MPER) for all scenarios with each park network. An increase in MPER indicates a loss of connectivity across the network, while a decrease in MPER indicates improved connectivity. We calculated change in MPER (ΔMPER) by taking the difference of MPER in a given scenario and MPER in the previous scenario to measure the effect size of adding new developments or protected areas. ΔMPER was calculated separately for development and protected area scenarios.

## Number of sentinel nodes required

*Koen et al. (2014)* found that for modelling omnidirectional connectivity, which uses source and destination nodes around the perimeter of a study area, a minimum of 20 nodes was required to accurately estimate current density throughout the study area using pairwise Circuitscape analysis. The authors showed that above this threshold, correlations between current density estimates and mean current density across the landscape reaches an asymptote, indicating additional nodes do not improve the connectivity model. We followed a similar approach to determine how many sentinel nodes are required to accurately represent the full park network.

To conduct this analysis, we used a ~4,200 km$^2$ study area containing ~100,000 pixels. This smaller study area was within the 57,000 km$^2$ landscape used in the above sensitivity analysis and was chosen to balance having a large enough landscape, while reducing computational demand. Using the 300-m cost surface of *Pither et al. (2023)*, we randomly selected 60 locations in the study area and assigned points to these locations which we assumed to represent the true protected area network. We then added 300-m buffers to these 60 spatial points, rasterized the buffered areas to a 300-m resolution, and reclassified as the lowest cost (0.1) to represent our full park network. We used simulated parks here to reduce any potential influence of park size and shape on the estimates of current density. Following the methods of *Koen et al. (2014)*, we connected randomly selected combinations of park nodes starting with two pairs and sequentially increasing until we reached the full 60 node complement, resulting in 59 different cumulative current density maps. We measured mean current density for all maps as well as the correlation between values of each map (2–59 nodes) with values of the full network (60 nodes). A high correlation between values of a given map and the full map suggests that the spatial configuration of current density estimates is similar (*Koen et al., 2014*). In addition, we calculated and compared mean pairwise effective resistance for each iteration. We note that each iteration used a random draw of nodes from the full network, and thus each estimate of current density and mean pairwise effective resistance is independent.

## Case study: assessing the current state of connectivity for Ontario's protected area network

We tested the sentinel node method on a larger landscape using the entire province of Ontario as a study area. Ontario covers an area of ~1.07 million km$^2$ and contains >1,400 protected areas including provincial parks, federal parks, and private protected areas. Using a sentinel node analysis would not only help to reduce computational demands, but would also provide a baseline evaluation of the state of connectivity for the Ontario protected area network and help inform future protected areas planning and design. For our provincial connectivity analysis, we used the 300-m resolution cost surface produced by *Pither et al. (2023)* cropped to the extent of Ontario, plus a ~325-km wide buffer to reduce edge effects (*Koen et al., 2010*, *2014*). For this province-wide analysis we assigned the lowest cost (0.1) to natural areas within parks and kept all other underlying landcover costs the same. Here, we make the assumption that natural lands within parks are less costly for animals to move through than natural lands outside of parks, while other landcover features within parks, such as roads, lakes, and built-up areas, remain costly.

Using our node analysis, we determined that 50 sentinel nodes would provide an accurate estimate of connectivity for the full park network. To initially select nodes, we limited the choice to only parks classified as 'Provincial Parks', since the analysis occurs within provincial boundaries and the province has jurisdiction over these parks; however, we integrated all classifications of protected areas (~1,423 PAs excluding protected portions of waterbodies) within the province into the cost surface. We also excluded parks with an area smaller than the size of a pixel (*i.e.*, 300 m × 300 m) and parks located on islands from selection as a node. This resulted in a pool of 314 provincial parks from which to select 50 sentinel node parks. To ensure an even distribution of nodes across the province, we divided the province into three zones: northeast, northwest, and south. To select the 50 park nodes, we used a stratified, random selection procedure with a set minimum proximity of 100 km between park centroids. Additionally, we specified that of the 50 parks, 20 each should be drawn from both the northeast and northwest zones and 10 from the south zone, which is roughly in proportion to the size of each zone. This selection procedure was implemented using the package *spatialEco* (v1.3-6; *Evans & Murphy, 2021*) in R (v4.2.2; *R Core Team, 2022*). Due to the number and proximity of parks within each zone, the selection algorithm converged on 45 parks representing 19, 14, and 12 parks within the northwest, northeast, and south zones, respectively. The remaining five parks were selected manually while ensuring sufficient distance from other parks. We used park centroids as node locations, provided centroids fell within a low cost (0.1) pixel. If the centroid fell within a higher cost pixel (1–1,000), we adjusted the node location to a directly adjacent or the nearest, lowest cost (0.1) pixel. If a park had no pixels with a cost of 0.1, the node was placed in a pixel with the next available, lowest cost. Similar methods for node placement were used by *Barnett & Belote (2021)*.

To map current density, which is an estimate of the probability of animal movement, we used pairwise mode Circuitscape in Julia with our input cost surface and 50 sentinel park nodes. Pairwise Circuitscape generates a cumulative current density map as an output by

 

connecting and passing current between all possible pairs of nodes (1,225 pairs for 50 nodes). The resulting current density map can be used to identify important animal movement pathways between protected areas. We sampled 1,000 random locations to compare our estimate of current density across Ontario to a recent estimate made by *Pither et al. (2023)* at the same extent and resolution and using the same input cost surface, but using different methods of circuit theory modelling (*i.e.*, sentinel nodes *versus* omnidirectional). We were interested to know how similar or different current density estimates produced using these two methods were. Circuitscape also calculates pairwise effective resistance, which is a measure of cumulative resistance between connected nodes. The more potential pathways between nodes, the lower the pairwise effective resistance and thus, greater connectivity. We calculated the mean pairwise effective resistance (MPER) across the 50 sentinel nodes as a measure of overall network connectivity.

### Node isolation

To measure isolation of individual node parks, we calculated the mean pairwise effective resistance between each sentinel node and all other nodes. This node isolation metric is similar to the PAI index used by *Brennan et al. (2022)* to measure the degree of isolation of global protected areas. Mean node isolation values can help identify parks within the network that are least connected and thus highlights areas where conservation interventions are most likely to help improve overall network connectivity.

## RESULTS

### Representation of a full network and sensitivity to changes in the network

Within the small study area in eastern Ontario, current density values from maps produced with randomly selected subsets of park nodes were highly correlated with each other and with current density values of a full "true" park network (mean *rho* = 0.91, range = 0.83–0.97). Mean pairwise effective resistance (MPER) between sentinel nodes, as a connectivity indicator, was able to detect simulated land-use and landcover changes. Across all random scenarios and the true scenario, MPER increased with the addition of a high-cost development and increased further with addition of a second and third high-cost development (Table 2). MPER decreased, indicating enhanced connectivity, with the addition of a new, low-cost protected area across all scenarios and decreased more sharply with the addition of a second and third protected area (Table 2). In neither case was the new park a node in the analysis, so the benefit to connectivity was achieved through the addition of new lower cost areas to the landscape. Our estimate of ΔMPER showed that in all cases, adding a development increased MPER as expected, and adding a park decreased MPER. The magnitude of these effects depended on the size and location of the new park or development (Table 2).

### Number of sentinel nodes required

We determined that 50 sentinel nodes was sufficient to accurately depict current density estimates of a simulated park network. Spearman rank correlations between current

**Table 2** Mean pairwise effective resistance values (ohms) of three random sentinel node scenarios (20 nodes/scenario; R1, R2, and R3) and the full "true" park network (52 nodes).

| Scenario | No development | Single development | Two developments | Three developments | Single park | Two parks | Three parks | Mean ΔMPER (se) developments | Mean ΔMPER (se) parks |
|---|---|---|---|---|---|---|---|---|---|
| New feature size (km²) | – | 157 | 199 | 676 | 33 | 336 | 150 | – | – |
| Landscape context | – | Low-cost | Moderate-cost | High-cost | High-cost | Low-cost | Moderate-cost | – | – |
| R1 | 116.102 | 116.164 (0.062) | 116.623 (0.459) | 117.236 (0.613) | 116.101 (−0.001) | 116.078 (−0.023) | 116.076 (−0.002) | 0.378 (0.164) | −0.0087 (0.0072) |
| R2 | 139.772 | 139.809 (0.037) | 140.401 (0.592) | 140.635 (0.234) | 139.767 (−0.005) | 139.751 (−0.016) | 139.750 (−0.001) | 0.288 (0.162) | −0.0073 (0.0045) |
| R3 | 280.496 | 280.528 (0.032) | 280.960 (0.432) | 281.412 (0.452) | 280.490 (−0.006) | 280.474 (−0.016) | 280.472 (−0.002) | 0.305 (0.137) | −0.008 (0.0042) |
| Full | 181.642 | 181.685 (0.043) | 182.154 (0.469) | 182.569 (0.415) | 181.638 (−0.004) | 181.620 (−0.018) | 181.619 (−0.001) | 0.309 (0.134) | −0.0077 (0.0052) |

**Note:**
Mean pairwise effective resistance values were calculated for different development scenarios including (1) the original cost surface; (2) addition of a high-cost human development; (3) addition of a second high-cost development; (4) addition of a third high-cost development; (5) the addition of a new low-cost protected area; (6) addition of a second protected area; and (7) addition of a third protected area. Effective resistance is a measure of the cumulative cost of moving between a pair of nodes. Mean pairwise effective resistance is an indicator of overall network connectivity. Values in parentheses show change in MPER (ΔMPER) for a given scenario with the previous scenario where positive values represent an increase and negative values a decrease in MPER. ΔMPER was calculated separately for development and park scenarios.

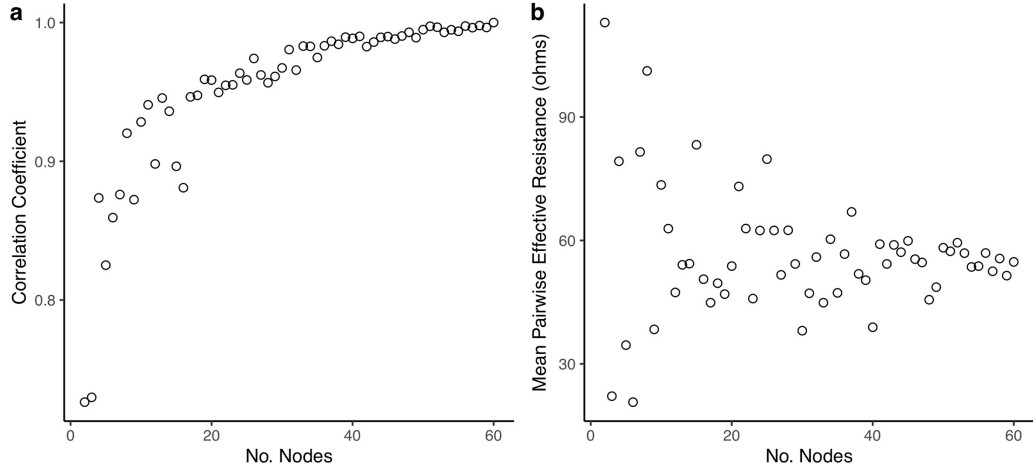

**Figure 1** **Effect of the number of nodes on current density and effective resistance.** (A) Spearman rank correlation coefficients of current density maps using increasing numbers of nodes compared to a current density map using all possible node connections. Number of nodes connected ranges from 2–60. All correlations are based on 1,000 randomly drawn current density values; (B) mean pairwise effective resistance values calculated for sentinel park node networks with increasing numbers of node pairs (range = 2–60 nodes).

density values from maps with increasing numbers of nodes compared to values from the full map (60 nodes) reached an asymptote at around 50 nodes (Fig. 1A; *rho* = 0.99 for the comparison between 50 and 60 nodes). Similarly, variation in mean pairwise effective resistance values was minimized at about 50 nodes (Fig. 1B).

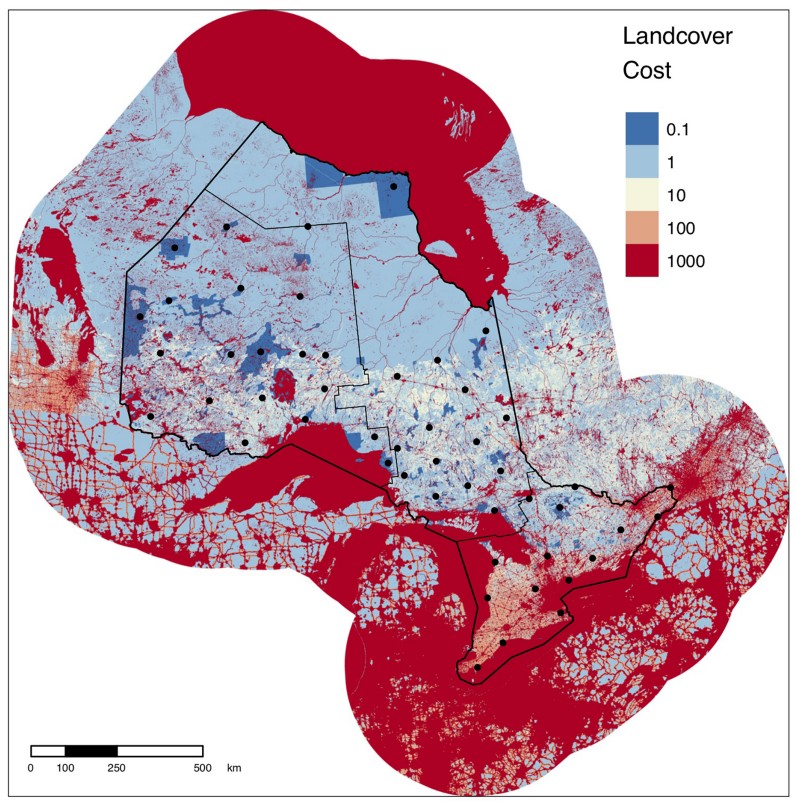

**Figure 2 Cost surface for Ontario with sentinel nodes.** Five cost values were assigned to landscape features based on the degree to which they facilitate or impede movement. Natural areas within protected area boundaries were assigned the lowest cost (0.1) under the assumption that they are less costly to move through than natural areas outside of protected areas. Solid black circles show the locations of the 50 sentinel nodes used to evaluate protected areas connectivity in Ontario. Solid black lines show the boundaries of the zones used to stratify the province for node selection. The modified cost surface was derived from the 300-m resolution cost surface of Canada from *Pither et al. (2023)*. The figure contains information licensed under the Open Government License–Canada.

## Case study: assessing the current state of connectivity for Ontario's protected area network

The final set of 50 sentinel nodes contained 20, 15, and 15 parks within the northwest, northeast, and south zones of Ontario, respectively (Fig. 2). The cumulative current density map for the full province displays varied patterns of current density between protected areas across the province (Fig. 3). For example, many areas of high current density (*i.e.*, pinch points) can be seen between parks connecting the south of the province to the north (Fig. 4B). This tendency for a pinching of current between south and north is magnified by the Great Lakes, as all current needs to pass to the north of the lakes. A lack of park-to-park connectivity is evident in the south of the province (Fig. 4C) where small protected areas are isolated by a matrix of human development. In contrast, areas of low, diffuse current density, where many potential movement pathways exist, can be found in the intact, low-cost northern regions of the province (Fig. 4A). There was a positive correlation (Spearman's *rho* = 0.76, $p < 0.001$) between values from our park-to-park current density

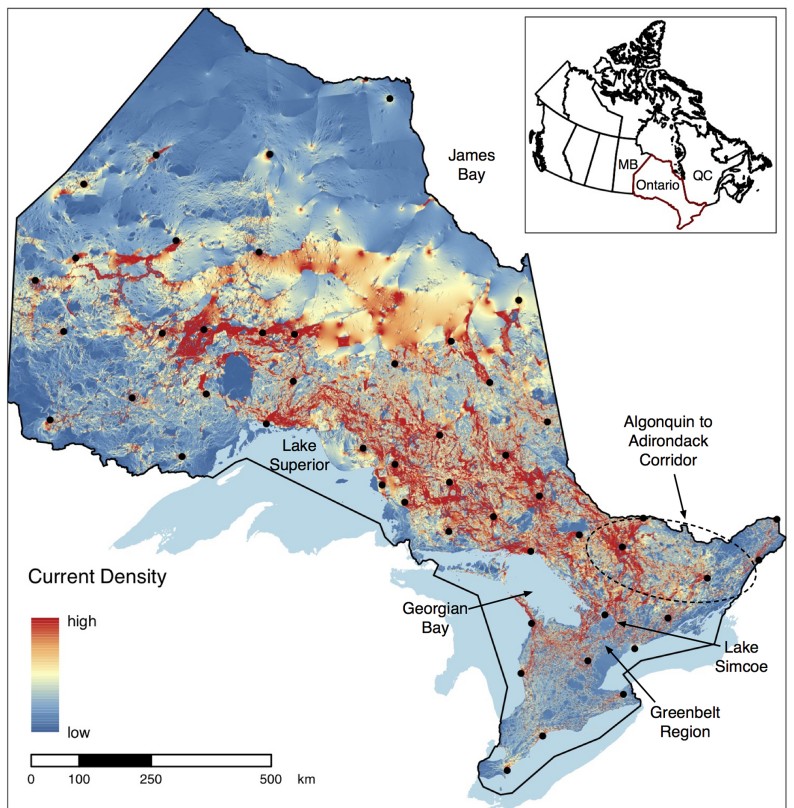

**Figure 3 Current density map displaying park-to-park connectivity for the protected areas network in Ontario.** Current density, measured in amperes (A), represents the probability of animal movement within a given pixel across the landscape. Sentinel node locations are depicted by black dots and the dashed oval shows the general extent of the Algonquin to Adirondack Corridor. The figure contains information licensed under the Open Government License–Canada.

map and values from the omnidirectional current density map produced by *Pither et al. (2023)*.

Mean (SE) pairwise effective resistance among the sentinel nodes was 83.07 (3.07) ohms. Pairwise effective resistance (also termed resistance distance) and Euclidean distances varied among pairs of nodes and there was a weak positive relationship between Euclidean distance and resistance distance (*rho* = 0.34, *p* < 0.001; Fig. 5). Node isolation values were positively skewed (Fig. 6A) and the highest isolation values were found in the south of the province (Figs. 4 and 6B; Table S1).

## DISCUSSION

We developed a new method using circuit theory to evaluate connectivity of a protected area network, and to establish an indicator of changing connectivity over time. In contrast to many other park-to-park connectivity models, we assigned a lower cost to protected areas in our cost surface and we use a fixed subset of protected areas as sentinel nodes to represent the full protected area network. Our method was sensitive to added developments, where the indicator increased as expected, and added parks, where the

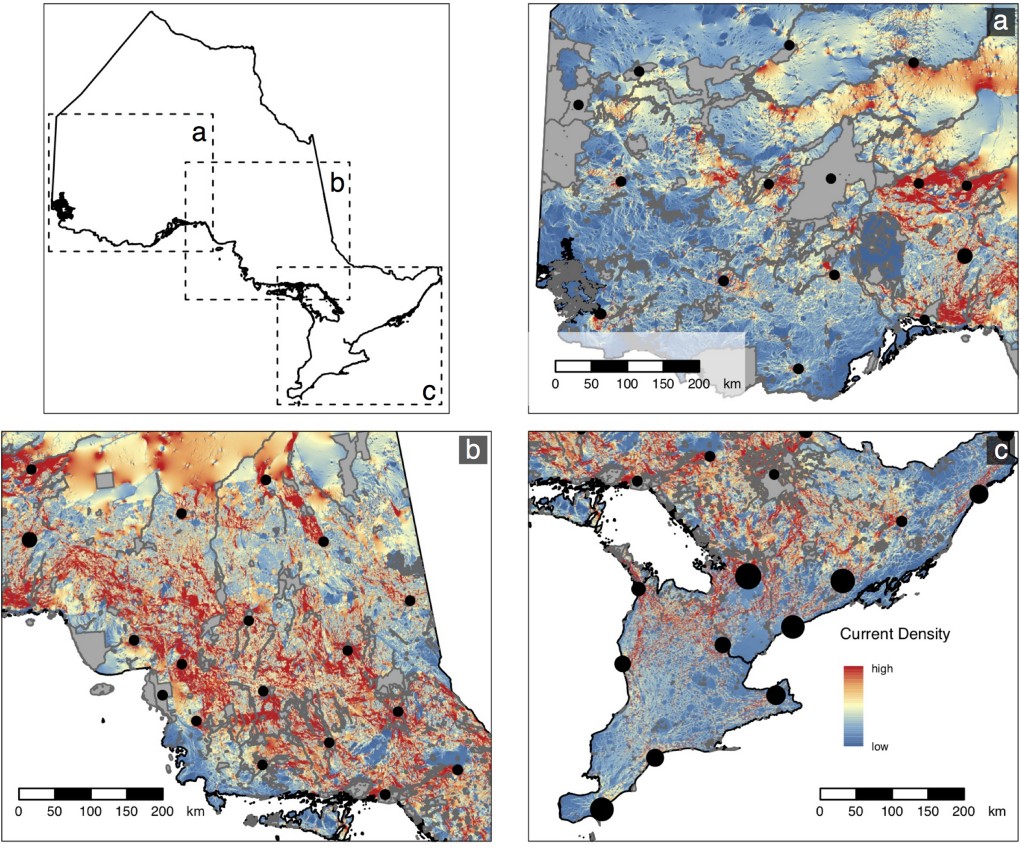

**Figure 4 Vignettes showing three areas that highlight differences in protected area connectivity across the provinces.** Regions displayed are (A) northwestern Ontario (B) central Ontario; and (C) southern Ontario. Grey polygons show protected and conserved areas and sentinel nodes are depicted by solid black circles. Size of the node is proportional to the degree of node isolation where larger nodes represent a higher degree of isolation from other nodes, and therefore parks, in the network. The figure contains information licensed under the Open Government License–Canada.

indicator decreased. Our method can be used to assess the current state of connectivity for a given protected area network and track changes in connectivity through time. As nations push to protect 30% of terrestrial lands by 2030 in well-connected protected area networks, the sentinel node method of measuring park-to-park connectivity provides a consistent and repeatable framework to incorporate connectivity into conservation strategies, and in particular, protected areas planning.

Our findings suggest that the sentinel node method is able to detect land cover and land-use changes. Specifically, we found that MPER was effective as a connectivity indicator to track changes in a protected area network. Under different simulated scenarios, MPER increased in response to added high-cost developments and decreased in response to addition of new protected areas in comparison to current estimates of connectivity. We consider that MPER meets many of the desirable properties of a connectivity indicator outlined by *Theobald et al. (2022)*. In particular, MPER incorporates interpatch distance (*i.e.*, resistance distance), reflects both within- and between-patch

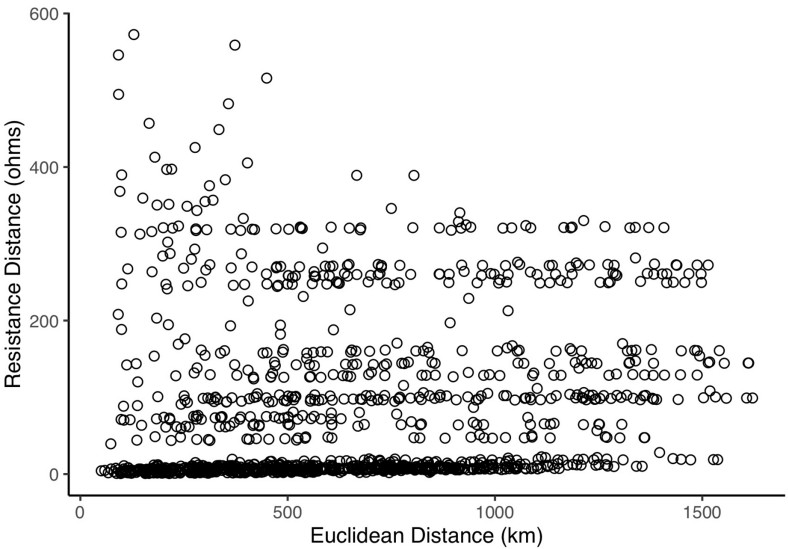

**Figure 5 Relationship between pairwise Euclidean distance and resistance distance of sentinel nodes.**

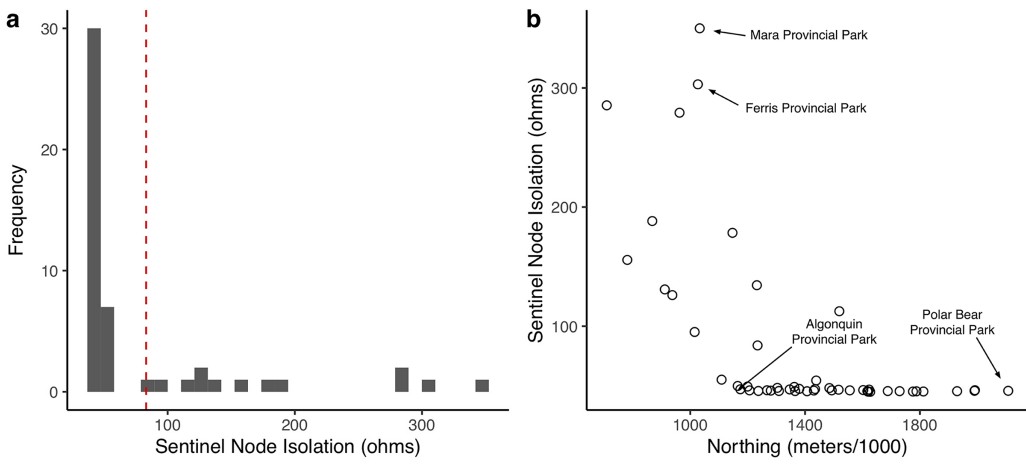

**Figure 6 Variation in node isolation values for sentinel nodes.** (A) Frequency distribution of sentinel node isolation values; higher values indicate a higher degree of isolation. The red dashed line shows the mean pairwise effective resistance for all 50 sentinel nodes (MPER = 83.07 ohms). (B) Relationship between sentinel node isolation and northing values.

connectivity, can be calculated at a high resolution (<1 km) with relative computational efficiency, and is simple to interpret. While values of MPER are not bound between zero and one as suggested by *Theobald et al. (2022)*, we maintain that MPER is still easily interpreted and unambiguous given it is bound between the range of cost values used (0.1 and 1,000 in our case).

In comparison to existing connectivity indicators (*e.g.*, ProNet and ProtConn), which may be more computationally efficient and simple to calculate, we think the MPER indicator brings added value in its repeatability over time and by its inclusion of landscape heterogeneity among parks. Similar to the findings of *Brennan et al. (2022)*, we would

expect to find contrasting results between MPER and other indicators due to differences in methodological approaches used to measure different aspects of connectivity (*e.g.*, structural *vs* functional). MPER and the PAI index are both generated using circuit theoretic connectivity models, however, a key difference is that MPER measures network-level connectivity while the PAI index measures node-level isolation. Another key difference is that we take a species-agnostic, naturalness approach to model connectivity for multiple species in general, while *Brennan et al. (2022)* modelled functional connectivity using species-specific data for 48 mammal species from across the globe. Further, the use of a static network of sentinel nodes sets MPER apart from the PAI index and other structural indicators (*e.g.*, ProNet and ProtConn). We consider MPER and our measure of node isolation are valuable additions to the suite of available indicators of connectivity and in combination with a second type of output from our analyses, current density maps that highlight areas with a high probability of animal movement, can help practitioners achieve progress towards conservation targets. Moreover, while we do not model functional connectivity as other indicators do (*e.g.*, PAI; *Brennan et al., 2022*), our naturalness approach to modelling connectivity is more cost effective and less time intensive than models requiring species-specific movement data. We think this is an important consideration for conservation practitioners who may be limited by time and funding. We suggest that future work could validate our model using independent wildlife data and could evaluate the alignment of our model with species-specific models of functional connectivity.

*Theobald et al. (2022)* highlight that a connectivity indicator should be computationally efficient and simple to re-estimate in the future. Circuit theory models have become increasingly popular for evaluating connectivity; however, with these resistance surface-based connectivity metrics, computational demand increases as the number of pixels in a cost surface increases and with an increasing number of nodes. Thus, it may become unrealistic for protected areas ecologists and planners to evaluate park-to-park connectivity using circuit theoretic models at a high resolution (<1 km) over a regional or national extent for an entire protected area network. *Koen et al. (2014)* showed that 20–30 nodes were needed around the perimeter of their study area to produce accurate estimates of connectivity using omnidirectional methods, while a higher number of nodes (~50 nodes) were needed when nodes were located within the study area. They suggested similar sensitivity analyses should be undertaken by other researchers to ensure sufficient sampling for their study. Following these recommendations, we determined that 50 nodes should also be sufficient to accurately represent Ontario's full protected area network when assessing park-to-park connectivity, however, we recommend other researchers test the generality of this finding in networks of varying structure. This could be evaluated using their own data and by following similar methods as outlined here and in *Koen et al. (2014)*. Where time may be a limiting factor, fewer iterations of nodes could be used, for example, producing current maps using increments of 10 nodes rather than increments of 1. Further research is needed to understand how the number of nodes required may change with the scale of the study, specifically the resolution of the data and extent of the study area. Overall, we suggest that using a subset of sentinel nodes reduces the computational
demand of running circuit theory models, allowing ecologists and planners to evaluate protected areas connectivity more efficiently while also providing a static network with which to track future connectivity.

We used our sentinel node method to evaluate connectivity of the full protected area network for the province of Ontario. The cumulative current density map shows differing patterns of current density within and between protected areas across the province (Figs. 3 and 4). As expected, connectivity is lower among protected areas in the south of the province, where small parks are isolated within a matrix of urban development and agriculture. Protection and restoration efforts (*e.g.*, tree-planting and grassland restoration) have a high potential to maintain and improve connectivity within this region. In contrast, protected areas in the northeast and northwest regions of the province are surrounded by a more intact landscape. Low to moderate current flow in these areas does not indicate low connectivity, but rather that many potential movement pathways exist and therefore current flow is more diffuse (*Marrotte et al., 2017*). While less affected by development, these areas are as important to conserve to maintain connectivity of these remaining intact landscapes, especially since protection of intact landscapes can provide more immediate and cost effective conservation benefits than restoration of degraded habitats (*Cook-Patton et al., 2021*). Many of the areas we identify align with previous omnidirectional models of connectivity (*Bowman & Cordes, 2015*; *Pither et al., 2023*), however, notable differences can be seen in the high current flowing through protected areas as a result of the low cost assigned to natural areas within park boundaries. While there are advantages to generating current density estimates using both methods, we believe our method is most appropriate in this case given our objective of evaluating park-to-park connectivity.

Importantly, the results of our case study highlight how our sentinel node method could be adopted by other regions (*e.g.*, provinces or states) and nations to evaluate connectivity of their own protected area networks and help inform decisions towards meeting Global Biodiversity Framework (GBF) targets. For example, the current density map output can be used to identify and prioritize critical connectivity areas for the expansion of well-connected protected areas (Target 3), to identify degraded lands where restoration work can enhance ecosystem function and connectivity (Target 2), and can be integrated with maps of other ecosystem services (*e.g.*, carbon storage; *Mitchell et al., 2021*; *Noon et al., 2021*; *Sothe et al., 2022*) to help identify areas important for nature-based solutions (Target 8; *e.g.*, *O'Brien et al., 2023*). Further, the resulting current density maps can complement existing GBF connectivity indicators (*e.g.*, ProNet; PARC), which can be used to determine how well connected a given protected area network is, but do not model potential movement pathways. Therefore, we believe current density maps produced using our sentinel node method can directly support existing indicators by providing a visualization of potential movement pathways, thus helping protected areas planners prioritize areas that will make the most meaningful contributions towards increasing protection of well-connected lands.

In addition to producing a current density map, which helps to identify areas important for maintaining and restoring connectivity, we also calculated MPER as an indicator of

connectivity for the Ontario protected area network. As this is the first time protected area connectivity has been evaluated in the province, this serves as a baseline evaluation of connectivity for which future estimates can be compared to track changes in connectivity through time. *Brennan et al. (2022)* found that a combination of restoration and protected area expansion maximized reductions in their PAI index, thus improving connectivity. Similarly, we found that addition of protected areas could decrease MPER of our park network. Our results and previous work suggests that land-use planners and conservation practitioners could maintain and enhance connectivity of protected area networks through a combination of protected area expansion, improved land management practices, and restoration efforts. Expanding current protected areas and enhancing connectivity between protected areas can help make significant contributions to global targets to conserve biodiversity by creating ecological networks of core areas and corridors (Target 3; *Convention on Biological Diversity, 2022*; *Hilty et al., 2020*). This will be especially critical in the face of climate change as currently suitable habitat in protected areas may become unsuitable in the future and species are required to shift their ranges to track suitable climates (*Parks et al., 2023*; *Schloss et al., 2022*). We suggest that future work should more rigorously test the sentinel node method by examining the response of MPER to protected area size and configuration and different conservation interventions (*e.g.*, addition of corridors).

We note that a few issues arise when attempting to re-evaluate a connectivity indicator for a protected area network over time that are important to recognize. For example, a protected area network is, and should be, dynamic over time as new protected areas are added, existing areas expanded, or new management practices implemented. These changes in the network across space and time impact the consistency of future estimates; however, the use of a static, unchanging set of sentinel nodes as we present here provides a repeatable framework to measure connectivity of an ever-changing network. Second, future changes in the network can become conflated with changes in the availability of new land cover data and technology or methods used to collect these data, which are likely to arise over the next few years. To reduce the likelihood of these issues affecting future assessments of connectivity, we suggest that (1) cost surface parameterization should follow the same cost ranking scheme (see *Pither et al., 2023*); and (2) the resolution of future analyses should match that of previous analyses, which may require aggregating new, higher resolution data to coarser resolutions (in this case 300 m). While we believe that the need to develop methods to monitor connectivity over time is ultimately more important, we acknowledge that the above steps are not a perfect solution to addressing issues that could arise from improved future landcover data and researchers should be aware of potential issues when comparing current and future estimates of connectivity.

The mean pairwise effective resistance, as an indicator of connectivity, was found to be 83.07 ± 3.07 (mean ± se) for the Ontario protected area network. This is a baseline estimate of the proposed connectivity indicator, which will be informative to compare over time to future assessments of the network. Similar to *Brennan et al. (2022)* PAI index, we believe estimates of MPER could also be used to compare networks across different regions (*e.g.*, provinces/states or countries) provided the same number of sentinel nodes, same cost

ranks, and same resolution (*i.e.*, 300 m) are used. Additionally, more fine-scale analyses could be undertaken on subsections of the province (*e.g.*, ecozones or ecoregions) to evaluate how certain regions contribute to connectivity across the whole province. For now, we can use our analysis to identify that there is spatial variation across the province in the degree of connectivity, and suggest that to enhance connectivity of the protected area network, a target would be to reduce MPER over time (*i.e.*, shift the estimate towards the lower range of cost values). This can be accomplished by reducing costs on the map in a variety of ways. Perhaps the most efficient way to reduce MPER would be to add protected areas in or near anthropogenically developed landscapes. We found that adding a larger protected area did not always have a greater effect on reducing MPER, demonstrating that in the right location, even small parks can be effective at enhancing connectivity. Evaluating alternative strategies for reducing MPER through simulation will undoubtedly be a useful strategy for landscape managers. With Canada's national targets to increase protection of terrestrial lands to 25% by 2025 and 30% by 2030, we suggest that it would be reasonable for connectivity of the Ontario protected area network to be re-evaluated in conjunction with these targets to track progress.

In addition to the mean pairwise effective resistance, we also calculated isolation values for individual sentinel nodes, which is comparable to the PAI index used by *Brennan et al. (2022)*. While it would be useful to calculate node isolation for all protected areas in the network, doing so would require re-running the analysis with all parks as nodes, which would negate one of the reasons for using sentinel nodes (*i.e.*, computational efficiency). Our node isolation index confirms that there is spatial variation in connectivity across the province with the highest degree of node isolation occurring in the south. Further, estimates of node isolation help to distinguish between nodes occurring in relatively intact, natural landscapes and those in heavily human modified landscapes. Both areas can have low current density values, but the latter will have a high isolation value (*i.e.*, high degree of isolation), while the former will have a low isolation value owing to many potential movement pathways throughout intact natural landscapes. We suggest that this index of node isolation helps to get at more fine-scale patterns and could be used to identify individual nodes where restoration work or other landscape management within and around a given protected area would help to improve overall network connectivity.

The recognized importance of connectivity for biodiversity is evident from the adoption of a new ambitious global biodiversity agreement including increased targets to protect 30% of terrestrial lands in ecologically representative and well-connected networks of protected areas. Key to achieving this target is the ability to measure connectivity and to track progress. We present a method through which government and non-government organizations can evaluate connectivity of protected area networks over time within a repeatable framework. This will allow connectivity to be better incorporated into protected areas planning, which is vital to creating functional networks of protected areas.

## ACKNOWLEDGEMENTS

Thank you to additional members of the Ontario Parks connectivity working group Karen Hartley, Louis Chora and Amanda Schroeder for advice. We also thank Richard Pither,

Angela Brennan, and Kristen Hirsh-Pearson for collaboration on related work and Jochen Jaeger for the insightful discussion. Special thanks to Richard Schuster and an anonymous reviewer who helped to improve this manuscript.

### Funding
Financial support for this study was provided by the Ontario Ministry of Environment, Conservation, and Parks for Paul O'Brien and Natasha Carr, and by the Ontario Ministry of Natural Resources and Forestry for Jeff Bowman. The funders had no role in study design, data collection and analysis, decision to publish, or preparation of the manuscript.

### Grant Disclosures
The following grant information was disclosed by the authors:
Ontario Ministry of Environment, Conservation, and Parks.
Ontario Ministry of Natural Resources and Forestry.

### Competing Interests
The authors declare that they have no competing interests.

### Author Contributions
- Paul O'Brien conceived and designed the experiments, performed the experiments, analyzed the data, prepared figures and/or tables, authored or reviewed drafts of the article, and approved the final draft.
- Natasha Carr conceived and designed the experiments, authored or reviewed drafts of the article, and approved the final draft.
- Jeff Bowman conceived and designed the experiments, authored or reviewed drafts of the article, and approved the final draft.

### Data Availability
The data is available at Figshare: O'Brien, Paul; Carr, Natasha; Bowman, Jeff (2023). Using sentinel nodes to evaluate changing connectivity in a protected area network. figshare. Dataset. https://doi.org/10.6084/m9.figshare.22751909.v1

### Supplemental Information
Supplemental information for this article can be found online at http://dx.doi.org/10.7717/peerj.16333#supplemental-information.

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
