# Peer review of "Using sentinel nodes to evaluate changing connectivity in a protected area network"

_PeerJ, doi:10.7717/peerj.16333_

## Round 0.1 · original submission · Major Revisions

Dear authors,

Many thanks for your manuscript submission. We have received two reviews and I believe the paper will be strengthened by including the changes and incorporating the comments mentioned by both reviewers. I look forward to receiving the revised version.

Reviewer 1 ·

Basic reporting

The manuscript contains a link to input and output data; however, it seems like it was not yet active.

Experimental design

no comment

Validity of the findings

no comment

Additional comments

The manuscript builds on a currently available tool (CircuitScape) to develop a new, elegant connectivity indicator, the Mean Pairwise Effective Resistance (MPER) between Sentinel Nodes with the goal of monitoring changes in the connectivity of protected area network as land use changes and new protected areas are established. The authors show that it performs as desired: the indicator value increases as connectivity decreases and vice versa.

It would be helpful for readers to receive more information about the strengths and weaknesses of this new indicator. A comparison to existing PA network connectivity indicators, especially those included in the Kunming-Montreal Global Biodiversity Monitoring Framework, would also be useful.
A strength that I am seeing, in addition to maintaining a set of sentinel nodes over time, is that it is advantageous to have an indicator that can easily be calculated with existing, widely used tools.
A weakness may be that with the selection of the sentinel nodes having a strong effect on the MPER values (Table 2), MPER is useful only for comparison between different time points of a study region, but not, for example, between countries. It also does not allow to set general connectivity targets for countries (e.g., connect x% of the PA system). Because the indicator is framed in the context of the Kunming-Montreal Global Biodiversity Framework, I would like to see a discussion of how this new indicator addresses or does not address countries’ GBF monitoring needs.
Below are additional line-specific comments:

Line 76: I suggest deleting ‘the’ in front of ‘tools’.
Line 81: I suggest deleting ‘the’ in front of ‘methods’.
Lines 87-90: I suggest also talking about input data for Circuitscape following this sentence.
Line 91: move 'measured in amperes' to after 'current density'.
Lines 195-198: I like this approach!
Lines 213-214: Can you please talk more (here and/or in the discussion) about internal park connectivity? Assigning all park grid cells a resistance value of 0.1 makes the assumption that connectivity within a park is perfect, disregarding potential fragmentation effects of busy park roads, developed areas, etc., correct?
Lines 321-322: Please state here why you compared your current estimate to Pither et al.’s.
Lines 323-324: Repetitive (see above)
Lines 391-393: Please also discuss which desirable properties MPER doesn't meet, e.g., bounded and linear values and why that matters (or not).
Line 395: Would it be computationally feasible to run MPER for an entire large country (e.g., Canada, Russia, China)?
Lines 398-399: This sentence makes it sound like MPER measures functional connectivity, which, if it does not as applied in this paper.
Lines 407-409: A paper you are citing (Theobald et al. 2022) developed a connectivity metric designed to do just that: ‘evaluate park-to-park connectivity at a high resolution (<1km) over a regional or national extent for an entire protected areas network’
Line 409: I suggest saying protected area network (not ‘areas’)
Line 421: Please refer to Figure 3.
Lines 447-451: This sentence does not address which of the models, your park-to park model or Omniscape models, may be better – which is what the previous sentence seemed to line up. With respect to climate change, both models may be useful to consider range shifts?
Lines 487-489: I suggest also mentioning ecological corridors and referencing the IUCN connectivity guidelines (Hilty et al. 2020) here.
Line 518: I assume you are referring to Canada’s national targets?
Lines 529-531: Please add information regarding what it would take to calculate the index of node isolation for all PAs, not just the sentinel nodes.
Figure 3: For readers not familiar with Canada’s provinces, please label Manitoba, because you are referring to it in the text. Also, I suggest replacing ‘amps’ with ‘ampere’.
Figures 3, 4: It would be helpful to see the sentinel nodes in these maps.
Figure 4: It would be helpful if you stated the MPER for each vignette in the figure.
Figure 5: You state that 'MPER incorporates interpatch distance (i.e., resistance distance)'. I may not be understanding it correctly, but can you please explain how the resistance distance can by 0 when the Euclidean distance is much higher (up to 1000km or so)?

·

Basic reporting

Thank you very much for the opportunity to review manuscript peerj-85480, “Using sentinel nodes to evaluate changing connectivity in a protected areas network”.

From the title and abstract it is not clear that connectivity is evaluated across the network only and not for individual nodes. If I understand the manuscript correctly this is an important point that needs to be made. Otherwise I am left wondering: why the additional approach/metric? How does this compare to the PAI index and what about the ProNet approach recently introduced.

The sentinel node generation seems a bit arbitrary to me. How do you propose a conservation practitioner determine their sentinel node network? What does it mean if we don’t take changes in the network over time into account?

I am curious about the leading sentence of a paragraph that you have on line 80:
“The rarity of connectivity being integrated into protected areas planning is not due to a
lack of the methods, as a wide variety of techniques exist for assessing protected area
connectivity.”
I was hoping that the paragraph eventually would tell me what could explain the rarity of connectivity being integrated into planning, but what followed was an introduction of methods. I would recommend to either i) describe what might be the cause of the phenomenon, or ii) probably more appropriate here to change the first sentence a bit to make clear that this paragraph will introduce methods and not point out why we see this phenomenon.

The last paragraph of the Introduction is very methods heavy and I would recommend shortening the paragraph to fit the “Introduction” style and moving any of the more methods specific parts into the methods section. As a reader I want to get a general sense of what you did here, not specifics about datasets (e.g. cost surface used).

Cost surface. Could you please elaborate with protected areas were set to low cost (1) across a protected area or were large water bodies and mountains still kept at high cost values? Either way, please justify your reasoning what cost adjustments please.

Random selection of sentinel nodes: There needs to be a lot more justification presented for using this approach. From the text in Methods (line 200++) I am not at all convinced that this is a reasonable approach.

Line 226 ++ (Representation of a full network and sensitivity to changes in the network) Now things become a bit clearer in terms of approach. I would highly recommend to let the reader know about ‘connectivity of the full park network’ early on (see comment about abstract above). I would recommend adding more replicates for the two scenarios. As a reader I am currently left wondering if the results you find are potentially based on the placement of the set of developed or protected areas. In other words, how sensitive are the results to your placement of developed and protected areas?

Line 257 (Number of sentinel nodes required). What’s your justification to conduct this analysis and the analysis described in the previous paragraph on two separate landscapes? Please justify what you did not investigate both representation and number of nodes on the same landscape.

Case study: You identified 314 provincial parks as node candidates and selected 50 of them. I am not sure about computational expense of running your approach with all 314 parks included, but I would be curious how different the ‘full’ result would be from the sentinel approach.

Discussion: What I would be curious about is a discussion of comparing networks with your MPER metric. Say, we are interested in comparing MPER scores of Ontario with MPER scores of Quebec. I don’t think that would work at this point, but I would be very curious about this and clarification on whether or not this is possible.

Line 395: “We plan to conduct more simulations to test the sensitivity of the sentinel node method, which would help in determining how well MPER, as a connectivity indicator, fits other properties outlined by Theobald et al. (2022).” I am wondering why this is not part of the current manuscript? If this not already underway, I would recommend removing this sentence.

Line 413. A generalization to state 50 nodes should be sufficient based on this current study is too broad a statement to be made here I think.

I think discussing the Ontario case study across 3 paragraphs of the discussion section is too much information in this manuscript. I would recommend you focus more on the methodological advancement you are making and less so on the specifics of the Ontario case study. From a readers’ perspective its currently not entirely clear whether this is a ‘methods paper’ or an ‘Ontario paper’. Methods and Results suggest methods, but Discussion focuses on Ontario a lot.

Line 501++. I am not sure that following the same cost ranking scheme and keeping resolution consistent will help much in terms of keeping analyses consistent across time. Thinking about the changes in landcover datasets available to us over the last 10-15 years I would think that there could be substantial change in landcover information over the next 10 years as well, which would make comparing current results to future results in MPER values challenging.

What are your thoughts on a MPER score for the entire network but also linking that to something more finer scaled like parts of Ontario? Is there a way that MPER could be used to compare between places (e.g. Ontario and Quebec) as well as sub-sections of say Ontario? Could it work to attribute certain parts of the MPER metric to certain regions? The way you describe the metric, I would say that should be possible. Maybe something to consider. I think having a hierarchy of MPER scores would be very interesting for a lot of people.

Finally, I would like to include a request to the authors to make their data and code as available as reasonable. I’m a big believer in the FAIR (Findable, Accessible, Interoperable and Reusable) principle for scientific data management and stewardship (e.g. https://www.nature.com/articles/sdata201618), which is why I include a request like this in all my reviews. Both at the review process and once the paper is published this would help a lot for replicating a study and validating findings.


Best regards,
Richard Schuster, PhD

Director of Spatial Planning and Innovation, Nature Conservancy of Canada
Email: richard.schuster@natureconservancy.ca

Experimental design

-

Validity of the findings

-

Additional comments

-

---

## Round 0.2 · Major Revisions

Thank you for your submission. Unfortunately, it does not seem the prior reviewer suggestions were sufficiently addressed. Please do so as per this second review. I look forward to your re-submission.

·

Basic reporting

Thank you very much for the opportunity to review manuscript peerj-85480v2, “Using sentinel nodes to evaluate changing connectivity in a protected area network”.

I would like to thank the authors for the effort they have put into this revision. I did find it hard to evaluate the reviewer responses document as there were responses that did not make it clear whether a reviewer comment lead to any changes in text or if the authors ‘just’ argued their point in that document. I would appreciate it if that could be made clear. If its just to respond to reviewer comments in the rebuttal document I don’t really see the value of this, because I do provide comments as I think through what potential readers would be curious about. If the response to those comments is restricted to the rebuttal document readers will not be aware of the reasoning the authors use.

Internal park connectivity and low cost areas (response 9). I still think there is an issue there and I don’t think its appropriate to use a lower cost for natural land cover in protected areas (0.1) compared to outside (1). Why would connectivity in natural areas be higher in parks?

Response 13: Could you please elaborate further how your work compares to that of Brennan et al. (2022)? I think this is important for readers to understand.

Response 14: I don’t see how the ‘clarifying text at lines 186-188’ justifies that argument that MPER measures functional connectivity. I disagree that MPER is the end of the gradient to functional connectivity. I find the liberal use of functional connectivity concerning here. I think the argument of a random walker is very weak and the tendency to overinflate the applicability of some methods (not just this paper) as functional connectivity measures is misleading.

Response 15: Given that this is really a slight conceptual modification on published work (adding circuit theory models), I would very much like to see a comparison to the method that has been published on before. At this point I can’t tell what actually changes between approaches. I know one argument is that MPER can be user over time, but so can the other method. The Theobold et al. (2022) method also uses a cost surface in their connectivity calculations. Given that I really wonder how MPER differs from the other method. This is not meant as a criticism, I am actually very curious and would like to understand the differences better.

Response 36. Thank you for this response. What I can’t tell from the response is whether this discussion has made its way into the main text. I don’t see a reference to the main text here. I think it would be useful to include this in the main text so readers get the benefit of being aware of this.

Response 38. Thank you for making the change to the text. As you recommend others to check the assumption that 50 nodes are sufficient, how do you propose they do so? What is your guidance on how many nodes to try? Does this depend on the study area setup? What considerations need to be taken into account? At this point the statement is not sufficient to help others determine what to do. This is of particular interest to non-academics who don’t have the time to spend a lot of time exploring the correct number of nodes for their work.

Response 40. As with Response 36 I can’t tell if this has made its way into the main text. I am not arguing that we should wait for perfect landcover data, that’s not possible in my opinion. I think its important to share with readers that one of the caveats of this method that can be used over time is the change in input data we can anticipate will happen given what we have seen over the last 10-15 years. If this method is meant to be ‘future proof’ this is an important consideration to be made. Keeping nodes consistent through time is great, but if my underlying resistance surface data changes significantly over time readers need to be made aware of the potential consequences of this.

Response 41. Thank you for this response. As with previous responses I can’t tell from the response is whether this discussion has made its way into the main text. I don’t see a reference to the main text here. I think it would be useful to include this in the main text so readers get the benefit of being aware of this.

I repeat my request to the authors to make their data and code as available as reasonable. I’m a big believer in the FAIR (Findable, Accessible, Interoperable and Reusable) principle for scientific data management and stewardship (e.g. https://www.nature.com/articles/sdata201618), which is why I include a request like this in all my reviews. Both at the review process and once the paper is published this would help a lot for replicating a study and validating findings. The current link to Figshare is not available to view. At this point I can not evaluate the validity of the authors statement that data and other information are there. I think its important to allow reviewers to evaluate a study thoroughly by making data and analytical code available to them.


Best regards,
Richard Schuster, PhD

Director of Spatial Planning and Innovation, Nature Conservancy of Canada
Email: richard.schuster@natureconservancy.ca

Experimental design

-

Validity of the findings

-

Additional comments

-

---

## Round 0.3 · accepted · Accept

Thank you for addressing the reviewer comments. I think they have been adequately addressed and I can recommend the manuscript for publication.